# Deep Learning-Based ADHD and ADHD-RISK Classification Technology through the Recognition of Children’s Abnormal Behaviors during the Robot-Led ADHD Screening Game

**DOI:** 10.3390/s23010278

**Published:** 2022-12-27

**Authors:** Wonjun Lee, Sanghyub Lee, Deokwon Lee, Kooksung Jun, Dong Hyun Ahn, Mun Sang Kim

**Affiliations:** 1School of Integrated Technology, Gwangju Institute of Science and Technology, Gwangju 61005, Republic of Korea; 2Department of Psychiatry, Hanyang University Hospital, Seoul 04763, Republic of Korea

**Keywords:** deep learning, skeleton, robot, attention deficit hyperactivity disorder

## Abstract

Although attention deficit hyperactivity disorder (ADHD) in children is rising worldwide, fewer studies have focused on screening than on the treatment of ADHD. Most previous similar ADHD classification studies classified only ADHD and normal classes. However, medical professionals believe that better distinguishing the ADHD–RISK class will assist them socially and medically. We created a projection-based game in which we can see stimuli and responses to better understand children’s abnormal behavior. The developed screening game is divided into 11 stages. Children play five games. Each game is divided into waiting and game stages; thus, 10 stages are created, and the additional waiting stage includes an explanation stage where the robot waits while explaining the first game. Herein, we classified normal, ADHD–RISK, and ADHD using skeleton data obtained through games for ADHD screening of children and a bidirectional long short-term memory-based deep learning model. We verified the importance of each stage by passing the feature for each stage through the channel attention layer. Consequently, the final classification accuracy of the three classes was 98.15% using bi-directional LSTM with channel attention model. Additionally, the attention scores obtained through the channel attention layer indicated that the data in the latter part of the game are heavily involved in learning the ADHD–RISK case. These results imply that for ADHD–RISK, the game is repeated, and children’s attention decreases as they progress to the second half.

## 1. Introduction

Attention deficit hyperactivity disorder (ADHD) is a chronic disease characterized by distraction, hyperactivity, and impulsivity [1,2]. Early diagnosis and treatment in childhood are important because childhood ADHD can progress to adolescence, as well as adulthood, if untreated. Although early diagnosis and prompt treatment are important [3,4], the incidence of ADHD in children is increasing worldwide every year [5]. In the United States, the incidence of ADHD in children is approximately 8.7% of all children, and 5.3 million children have been diagnosed with ADHD from 2016–2019 [6]. This increase could be attributed to the new ADHD guidelines that resulted from the recently approved revision of the Diagnostic and Statistical Manual of Mental Disorder, Fifth Edition (DSM-5) [6]. As the incidence rate of ADHD increases, more research into its treatment is being conducted in various ways. The most widely used treatment method for ADHD so far is drug treatment. ADHD drug treatment is fast, and its efficacy has been verified [7,8]. However, as a result of the side effects of the drug, negative attitudes toward drug treatment are increasing. To avoid the side effects of drug treatment, non-drug treatment methods are being studied [9,10]. Methods using the Internet of Things, artificial intelligence, augmented reality, virtual reality, and robots are being studied as nonpharmacological ADHD treatment methods. Particularly, many methods for treating ADHD using robots are being developed [11]. Representative examples are NAO, Pepper [12], Silbot [13,14], Sanbot Elf [15], and Bioloid humanoid robot [16]. The aforementioned robots help children with ADHD through speech therapy development, motor skills therapy, traditional therapeutic improvement, attention and memory therapy, and so on [11]. Although there are various nonpharmacological treatment methods for ADHD using robots, research on ADHD diagnosis using robots is limited. Early diagnosis is essential for prompt treatment, but research on ADHD diagnosis using robots is still insufficient. However, studies on ADHD screening diagnosis using various sensors and deep learning or machine learning are in progress.

The first example is a method that uses the inertial measurement unit (IMU) and machine learning. After attaching two IMU sensors to the children’s ankles and waist, IMU data obtained while the children and their parents were consulted by a doctor for approximately 1 h were used as input data. In the case of the machine learning algorithm, a support vector machine (SVM) was used, and an accuracy of 95.12% was finally obtained [17]. The second example is a method of diagnosis that adopts the continuous performance test results currently used by doctors in hospitals for ADHD screening and diagnosis as input data and employs a machine learning algorithm. The above method obtained diagnostic accuracy of 87% and 86.3% using random forest and SVM, respectively [18]. The last example is a diagnostic method that uses electroencephalogram (EEG) data and functional magnetic resonance imaging (fMRI) results, both of which have been widely used in recent studies. ADHD diagnosis research is underway using EEG data as well as deep learning and machine learning algorithms. Among them, Tosun et al. achieved 92.2% accuracy in ADHD diagnosis results using long short-term memory (LSTM) [19]. Additionally, with the release of the Neuro Bureau ADHD-200 dataset, research on ADHD diagnosis using fMRI data is being actively conducted [20]. Many research teams are conducting research to improve the accuracy of ADHD diagnosis using the above dataset and deep learning or machine learning. Among them, Chen et al. achieved 88.1% accuracy in ADHD diagnosis using the SVM algorithm [21]. With the release of the excellent dataset known as the Neuro Bureau ADHD-200 dataset, it is expected that studies on the improvement of ADHD screening and diagnosis performance using deep learning or machine learning will continue in the future. Recent ADHD classification studies using various data sets, including the studies mentioned so far, are summarized in Table 1 below.

The abovementioned ADHD screening studies also have limitations. Since the method using the IMU simply measures the amount of exercise performed by children, there is insufficient information to perform various analyses. Studies using fMRI or EEG can be objectionable to children because hospitals use specialized equipment and children must go inside the MRI machine or have EEG electrodes placed on their heads.

Additionally, for data acquisition, children visit a hospital, in an unfamiliar environment, and participate in a data acquisition experiment using various sensors or testing equipment. If nonadult children are subjected to an experiment in an unfamiliar environment, they may not engage in their usual behavior. Because ADHD is diagnosed on the basis of problem behaviors observed while observing children’s usual behavior, the accuracy of diagnosis can be further improved by conducting an experiment in an environment in which children’s usual behavior can appear naturally [27].

To overcome this limitation, in this study, the data used were acquired through robot-led games in schools and children’s centers familiar to children. Furthermore, in this study, the robot-led ADHD screening game used was developed by ADHD specialists and child psychologists to naturally draw appropriate stimuli and responses from children through robots and games.

Moreover, children can participate in this experiment without any objection because of the content of games, robots, and the familiar environment for children.

The four methods mentioned above are currently the most studied methods for classifying ADHD using machine learning or deep learning. In the case of the IMU introduced first, children feel uncomfortable because they wear the IMU device on their body for a long time. However, unlike the IMU method, since this system allows children to play games for a short period of time without wearing equipment, relatively objective data can be obtained because they can focus only on the game while obtaining data, unlike the IMU method. In addition, in the case of EEG and fMRI data, data acquisition must be performed in a hospital and this requires very expensive and special equipment. However, in the case of this system, children’s data can be obtained with simple equipments in a child-friendly environment because data are obtained from a school familiar to children. Finally, the methods using questionnaires or test scores, which are currently used most often in hospitals, have the disadvantage of low objectivity. However, since the developed system uses data obtained while all children play the same game in the same environment, objective results can be extracted using objective data. All references mentioned in this paper classify only ADHD and normal groups. However, ADHD–RISK was classified in this study. In this study, unlike other papers, high accuracy was obtained by including the ADHD–RISK group as well as the ADHD and normal groups.

It is difficult for clinicians and professional teachers to identify the ADHD–RISK class, so it was rarely used in previous studies. However, ADHD–RISK is necessary for systematic ADHD screening. In this paper, the ADHD–RISK class was added to develop a systematic screening tool for clinicians and professional teachers.

The contributions of this paper are summarized below.


In this paper, ADHD, ADHD–RISK, and Normal groups were screened with high accuracy using skeleton data and deep learning obtained through relatively simple measurements through interesting simple games in a child-friendly natural environment without expert intervention.Unlike previous studies, the ADHD–RISK class, which is not easy for clinicians to distinguish, was added, and high classification accuracy was obtained.ADHD, ADHD–RISK, and Normal groups were classified with high accuracy using only skeleton data, which was previously not used for ADHD classification.When classifying ADHD–RISK children through the channel attention layer, it was verified that classification becomes more helpful toward the later part of the game.


The detailed structure of this paper is as follows. The introduction section introduces the existing methods for screening ADHD. The materials and methods section introduces the materials used in this study and the methods proposed. The results section describes the experimental results. The discussion section describes the discussion in this study. Finally, the conclusion section presents a conclusion.

## 2. Materials and Methods

### 2.1. Skeleton Acquisition System Using Robot and Five Depth Sensors

Figure 1 shows the overall diagram of the system for acquiring children’s skeleton data. Five depth sensors were used to collect skeleton data while children played a game created by child psychologists and clinicians. The depth sensor used is Microsoft’s Kinect Azure, a commercial product used in various studies on skeleton data acquisition. In a previous study, this research team conducted human tracking using five Kinect Azure and the skeleton data acquisition method using the skeleton merge algorithm acquired from five sensors. For more information, see reference [28]. Additionally, Robocare’s Silbot, a robot for overall game progress and game demonstration, was used. Lastly, LG’s beam projector (HU85LA) was used to display the game progress.

### 2.2. Robot-Led ADHD Screening Game for Children to Acquire Skeleton Data

Children play the game to collect skeleton data. The game can be divided into three main stages. When the game starts, the game screen is displayed on the floor using the projector. A total of five games are played, starting with the pre-explanation stage, in which the robot explains the overall game. Each of the five games is divided into two stages: a waiting stage in which the robot performs a game demonstration and a game stage in which children directly participate in the game after the robot game is over.

The robot-led game played by children is described as follows. Nine number plates are displayed on the floor using the projector. The robot randomly passes through the printed numeric keypad. At this time, the child memorizes the number board passed by the robot in the waiting area, and when the robot returns to its original position, the child must pass the number board passed by the robot in the same way according to the start signal given by the robot (Figure 2).

Furthermore, children must perform one more mission while passing the number board. While the children are playing the main game, characters appear on the screen, as shown in Figure 3. When the scarecrow, robot, and lion from the Wizard of Oz appear, the children stop moving and wave their hands, as shown in Figure 3a. However, if a witch appears, the children must stop moving and sit down, as shown in Figure 3b. After completing the intermediate mission, the children must continue playing the game to complete the remaining route movement. Finally, 11 types of input data are obtained, including preliminary stage data for each child, waiting stage data acquired for every five games, and two data points for each game stage. For more information about the game, see reference [27].

### 2.3. Data Acquisition Participants and Data Acquisition Methods

The children participating in the game were recruited from 8- to 13-year-old elementary school students from Seoul, Korea. We tried to reduce environmental variables as much as possible by playing the game in schools or child research centers near where children live, rather than in unfamiliar environments such as hospitals or research institutes. From 2019 to December 2021, skeleton data of children were obtained at three elementary schools and one child research center where children attend. All children who participated in the game had their parents’ consent, and they were all administered the DSM–ADHD scale, CBCL (Child Behavior Checklist), and K–ADHDS (Korean ADHD Diagnostic Scale). After seeing the results of the above three screening tests and the video of the children who played the game, four doctors then discussed and diagnosed each child’s assignment into normal, ADHD–RISK, or ADHD groups, and the results were used as the children’s label. As shown in Table 2, data from a total of 596 children were obtained. Among them, 349 were classified as normal, 181 had ADHD–RISK, and 66 had ADHD.

### 2.4. LSTM-Based Deep Learning Algorithm for Selective ADHD Screening Using Attention Layer

Skeleton data of children acquired through a robot-led ADHD screening game were used as input data. The input data were divided for each stage obtained from the total game. A total of 11 input data types were used: six waiting stage data, including the pre-explained stage, and five game stage data. Since each stage takes a different amount of time for each child to complete the game, a data structure was created based on the longest frame for each stage, and then all data were matched to the structure. In the case of data from a frame shorter than the reference data structure, the remaining frames were filled with 0.

A total of 32 skeleton data points were provided by Kinect Azure; however, in this study, 18 joints with excellent skeleton acquisition quality were used, as shown in Figure 4, based on the results of previous research conducted by this research team [28].

Figure 5 shows the deep learning model for ADHD classification using the input data after the data preprocessing process. The deep learning model used was bidirectional LSTM.

As shown in Figure 6, the bidirectional model uses two separate hidden layers to process data in the forward and reverse directions. The final output y is obtained by concatenating the hidden states f and b of the t-th forward LSTM cell and the t-1th backward LSTM cell. After passing through the bidirectional LSTM layer for each stage, the generated features are passed through the channel attention layer to improve performance and evaluate which stage has the greatest influence on learning for each stage.

In this study, an attention layer for the stage level was proposed to improve recognition performance. This was inspired by the channel level attention layer proposed in the reference paper [29]. First, the feature corresponding to each stage extracted from the bidirectional LSTM layer described above is concatenated to the channel level and used as an input. Second, the stage attention block proposed in this paper encodes each stage feature through max pooling and average pooling, and then it is delivered to the same feature shared network. It is composed of two fully connected layers, similar to channel attention in reference [29], and the number of features is reduced to 1/2 and then restored to the original number. According to reference [29], this structure improves the generalization performance of the recognition model. Third, the two extracted features are added element-wise and scored using a sigmoid that transforms the feature range from 0 to 1. Equation (1) expresses the above process.
(1)$$M_{c}\left(F\right)=\;\sigma \left(\mathrm{MLP}\left(\mathrm{AvgPool}\left(F\right)\right)+\;\mathrm{MLP}\left(\mathrm{MaxPool}\left(F\right)\right)\right)$$

That is, during the classification, the importance of features for each stage is randomized and derived. Finally, the attention score is multiplied for each stage of the input of the attention block, and weighting is applied according to the importance of the feature. The application of attention to each stage proposed in this paper can induce learning, allowing the recognition model to prioritize important stages. Moreover, the model not only can improve recognition performance but also provides an interpretation of which stage is an important stage in classifying ADHD based on the attention score applied during classification. Figure 7 illustrates the proposed stage attention block.

ReLU was used as the activation function, and the resulting reshaped feature was passed through the classification layer to classify normal, ADHD–RISK, and ADHD groups. Since the number of input data used varies, a weighted cross entropy loss function, as shown in Equation (2), is used to prevent overfitting and improve performance. *t_i_* is the truth label, *pi* is the Softmax probability, and *w_i_* is the weight of the loss function.
(2)$$\mathrm{Weighted}\;\mathrm{Cross}\;\mathrm{Entropy}\;\mathrm{Loss}\;=-\overset{\;C}{\underset{i=1}{\sum }}w_{i}t_{i}\log\left(pi\right)$$

In the case of weights, the inverse of the input data ratio was used, and for verification, the leave-one-person-out cross-validation method was used. The learning rate of the model used in the paper above is 0.001, the epoch is 20, the batch size is 50, and when four LSTM layers are used, the hidden size of LSTM is 128. Total number of parameters is 15,185,521.

## 3. Results

Normal, ADHD-RISK, and ADHD were classified using an LSTM-based deep learning algorithm and skeleton data of children acquired through a robot-led game for screening children’s ADHD. In this paper, the model was evaluated using accuracy, specificity, sensitivity, *F*1 score, *FNR*, *FPR*, and *FDR*. The formulas are as follows [30]:(3)$$Accuracy\;\left(Acc\right)=\;\frac{TP+TN}{TP+TN+FP+F\;N}$$
(4)$$Specificity\;\left(Sp\right)=\;\frac{TN}{TN+FP}$$
(5)$$Sensitivity\;\left(Se\right)=\;\frac{TP}{TP+FN}$$
(6)$$F1=\;\frac{2\;\times TP}{2\times \;TP+FP+FN}$$
(7)$$FNR=\;\frac{FN}{FN+TP}.$$
(8)$$FPR=\;\frac{FP}{FP+TN}\;$$
(9)$$FDR=\;\frac{FP}{FP+TP}\;$$

In the above equations, *TP* (true positive) represents the result of predicting the correct answer that is actually true as true; *FP* (false positive) is the result of predicting the correct answer that is actually false as true; *FN* (false negative) is the result of predicting that the correct answer that is actually true is false; and *TN* (true negative) is the result of predicting that the correct answer that is actually false is true.

Table 3 shows accuracy, sensitivity, specificity, *F*1 score, *FPR*, *FDR*, and *FNR* for each class. Accuracy, sensitivity, specificity, and *F*1 score for the normal class were 98.99%, 100%, 97.57%, and 99.14%, respectively; for the ADHD–RISK class they were 98.15%, 93.92%, 100%, and 96.86%, respectively; for the ADHD class they were 99.16%, 100%, 99.06%, and 96.35%, respectively. Also, *FPR*, *FDR*, and *FNR* for the normal class were 0.024, 0.016, and 0, respectively; for the ADHD–RISK class they were 0, 0, and 0.06, respectively; and for the ADHD class they were 0.009, 0.07, and 0, respectively. Finally, the classification accuracy of the three classes was 98.15%. In our previous study, which was conducted without using the channel attention layer, a classification accuracy of 97.81% was obtained for the classes.

Figure 8 shows the results of the ADHD classification confusion matrix. In the case of the normal class, all 349 normal children were classified as normal. In the case of the ADHD class, all 66 children with ADHD were classified as ADHD. However, in the case of the ADHD–RISK class, 170 of 181 ADHD–RISK children were classified as ADHD–RISK, and six and five children were classified into the normal and ADHD classes, respectively.

## 4. Discussion

In this study, an ADHD screening algorithm was developed using the skeleton data of children acquired through games for ADHD screening. Moreover, the channel attention layer was used to improve the performance of the algorithm by verifying which parts of the waiting stage and the game stage developed for the classification of normal, ADHD–RISK, and ADHD are helpful for ADHD screening and classification. To verify the significance of each stage, first, the attention score obtained through the channel attention layer was analyzed. Following the completion of learning, attention scores were generated for each stage for the 596 children who participated in this study. Table 4 shows the average of the generated attention score.

Consequently, no attention score showed a significant difference for each stage. Based on the above results, it was determined that all stages had a uniform influence on the learning outcomes. Second, the attention layer was analyzed for each class rather than an overall analysis. In this analysis, as shown in Table 5, the distribution rate of the top three attention scores out of a total of 11 attention scores for each stage of each class was confirmed.

Consequently, in the case of the normal and ADHD classes, no significant deviation in all stages was observed. However, in the case of the ADHD–RISK class, significant results were confirmed for both the waiting and game stages. In both the waiting and game stages, it was confirmed that the upper distribution rate of the attention score increased from the start of the third game.

To confirm the results more easily, the above results are shown in a graph, as shown in Figure 9. In the waiting stage, the distribution rate of the upper attention score was 5% until the second waiting stage, but it increased sharply from the third waiting stage, showing an average distribution rate of approximately 12%, as shown in Figure 9a. Similarly, in the game stage, the average distribution rate of the upper attention score was 5% up to the second game stage, but it increased to approximately 12% after the third game stage, as shown in Figure 9b. The above results indicate that children with ADHD–RISK played the game while focusing on the game at first but lost interest and could not concentrate on the game by the third time, as they became accustomed to the game. Additionally, in the case of the normal and ADHD groups, it can be inferred that the children had a similar appearance with no distinguishing characteristics in all stages. The doctors who proposed and designed the above game expected that the movements of children in each group would be different during waiting and gaming. Additionally, it was determined that the difference in motion would also be detected in the acquired skeleton data. When the children in each group were observed while playing the game at the time of data acquisition, the children in the normal group showed a state of focusing on the robot’s movements from the first stage to the last stage with almost no special unnecessary movements. On the other hand, children with ADHD did not concentrate on the robot from the first stage to the last stage and showed a lot of unnecessary movements while playing the game, such as sitting and standing up, shaking their legs, or shaking their arms. In the case of the two groups, the difference in movement during the game was clear. For this reason, as shown in Figure 9, it was found that there was no significant difference in attention scores by stage in all stages.

When the children in the ADHD–RISK group were observed while playing the game, it was confirmed that they played the game similarly to the normal group. For this reason, it is judged that it is not easy for doctors to distinguish between ADHD–RISK and the normal group. However, in the result of the attention score obtained through the channel attention layer, the characteristics of ADHD–RISK were found as shown in Figure 9. Based on the above results, it is expected that better screening and classification will be possible in the future if the game is designed to require more concentration as the game progresses for a more accurate classification of ADHD–RISK. Unlike previous ADHD studies, this study included the ADHD–RISK class, and it was determined that if this system is used in actual clinical trials in the future, it will be helpful for special education teachers and clinicians in diagnosing. However, in the case of ADHD–RISK, even when an actual human is discriminated against, there is a difference in diagnosis between doctors, making it difficult to accurately identify it. Similarly, the deep learning model proposed in this study accurately classified 100% of the normal and ADHD classes, but inaccurately classified 11 out of 181 people in the ADHD–RISK class. However, based on the results obtained through the attention layer, in the future, we think that the above problem can be overcome if the development of an improved classification algorithm is combined with the development of a game for ADHD screening that requires more concentration in the second half.

## 5. Conclusions

In this study, a deep learning algorithm based on skeleton data and LSTM acquired through games was developed for ADHD screening of children, and normal, ADHD–RISK, and ADHD class classification studies were conducted. The channel attention layer was used to improve the performance of the algorithm and to analyze the developed game. Using the final algorithm developed in this study, the classification accuracy of the three classes was 98.15%. Furthermore, by analyzing the attention score obtained through the channel attention layer, it was confirmed that in the case of ADHD–RISK, children’s concentration decreased toward the end of the game. If in the future we design a new game that requires more concentration in the second half based on the above results, it is expected that it will aid in ADHD screening by increasing the classification accuracy of the ADHD–RISK class, which has lower classification accuracy than other classes.

## Figures and Tables

**Figure 1 sensors-23-00278-f001:**
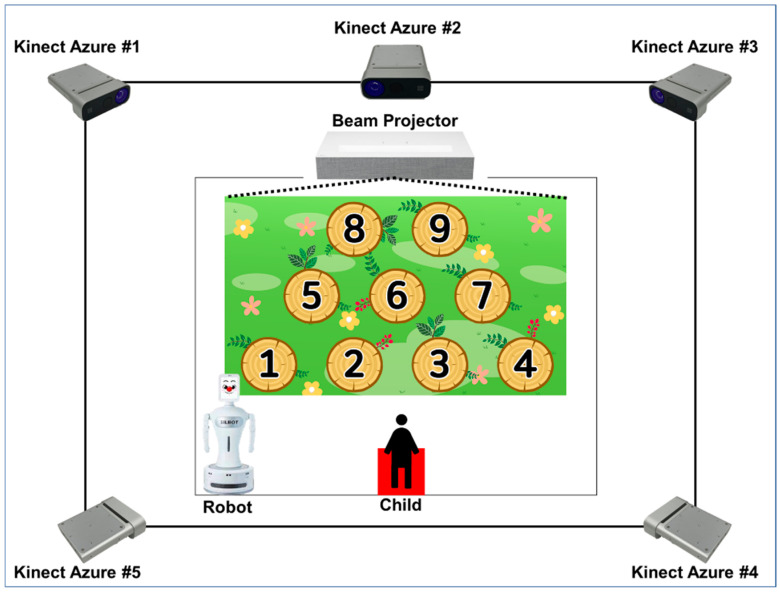
System for acquiring skeleton data for attention deficit hyperactivity disorder (ADHD) screening of children. Five Kinect Azure devices were used to acquire skeleton data and a beam projector and a robot were used to run the game.

**Figure 2 sensors-23-00278-f002:**
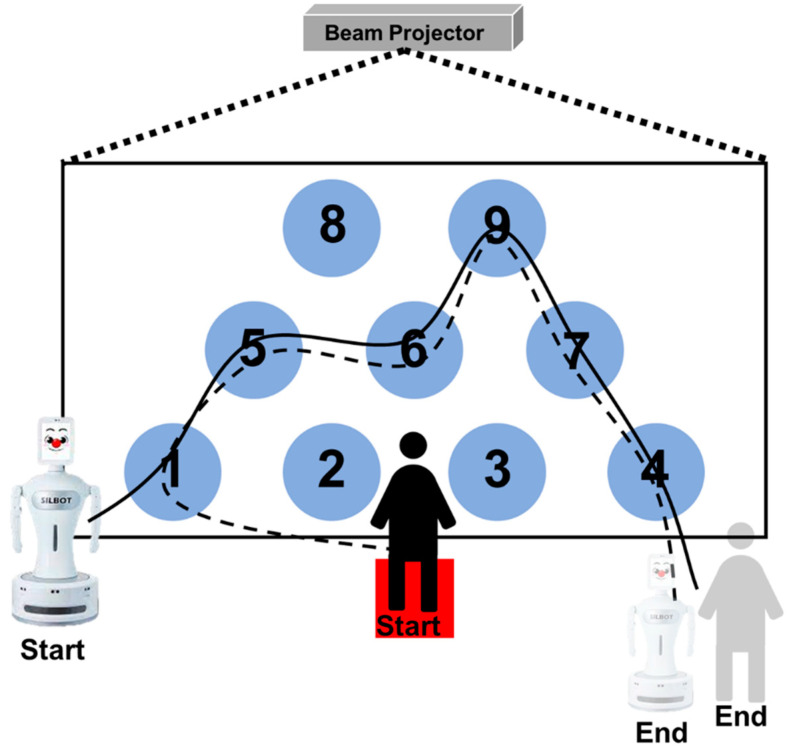
Game description for ADHD screening of children. A game in which the robot first passes the number board, the child remembers the number passed by the robot, and then follows the same path.

**Figure 3 sensors-23-00278-f003:**
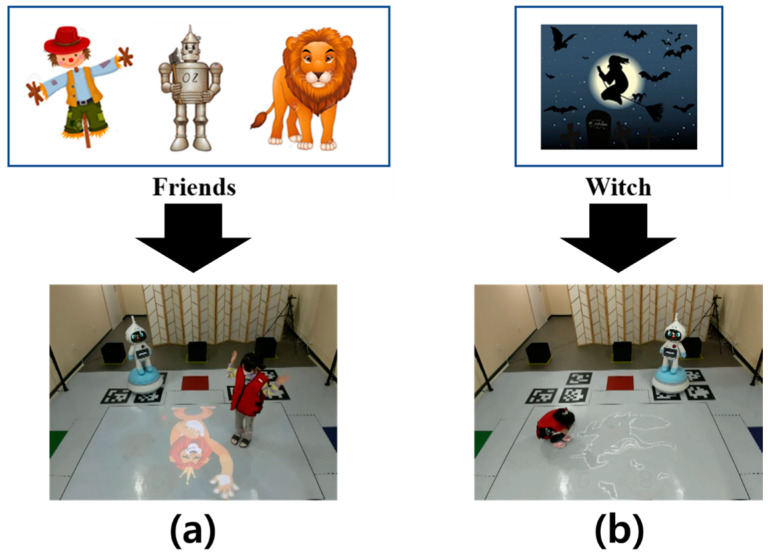
Description of the intermediate mission during the game that follows the path of the robot. (**a**) When the scarecrow, robot, and lion are printed on the floor, the child must wave their arms. (**b**) When the witch is printed on the floor, the child must sit down.

**Figure 4 sensors-23-00278-f004:**
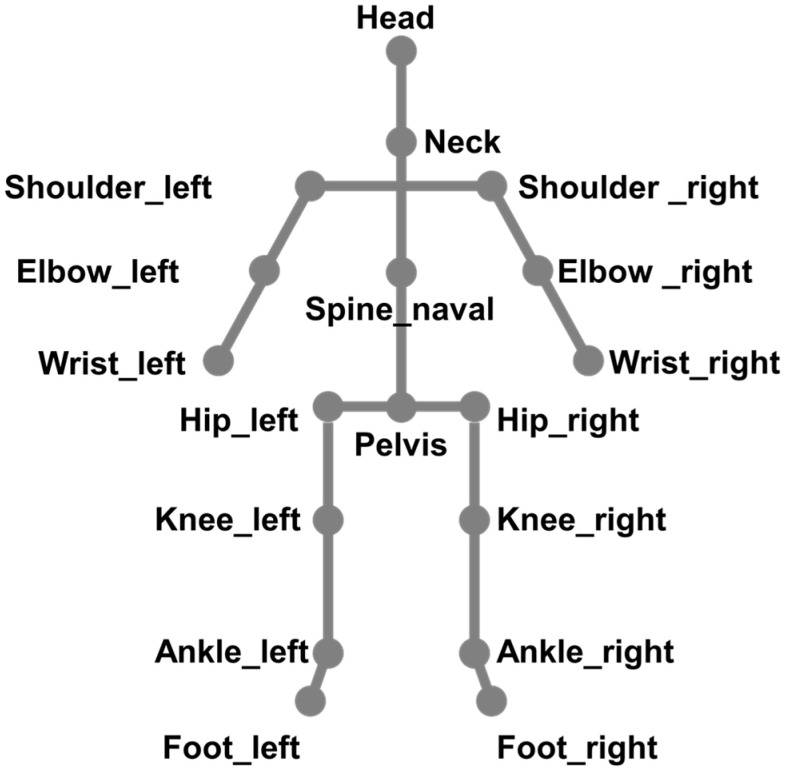
Eighteen types of skeleton data acquired and used for ADHD screening of children.

**Figure 5 sensors-23-00278-f005:**
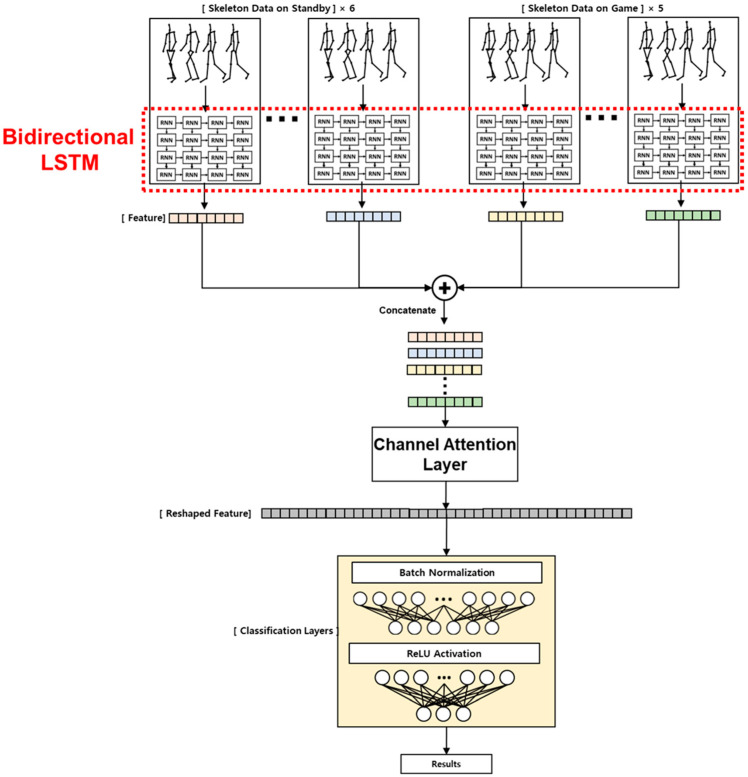
Deep learning model with an added attention layer based on bidirectional long short-term memory (LSTM) used for ADHD screening of children.

**Figure 6 sensors-23-00278-f006:**
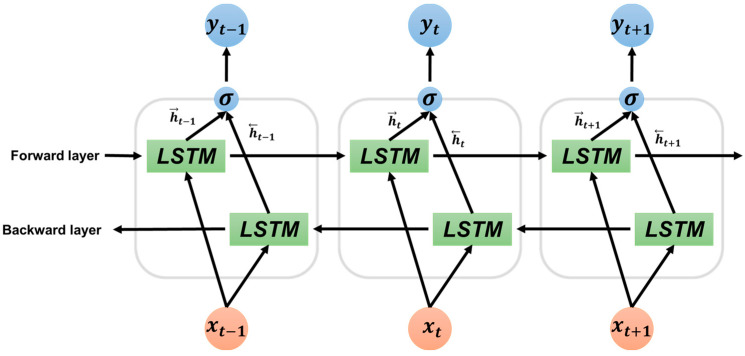
Structure and description of bidirectional LSTM used to improve model performance.

**Figure 7 sensors-23-00278-f007:**
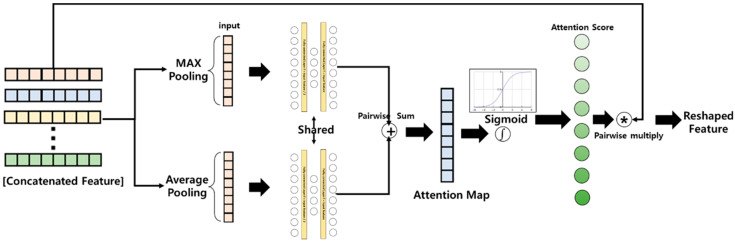
Channel attention layer designed to improve the performance of the model and the importance of each stage of the skeleton data acquired for ADHD screening of children.

**Figure 8 sensors-23-00278-f008:**
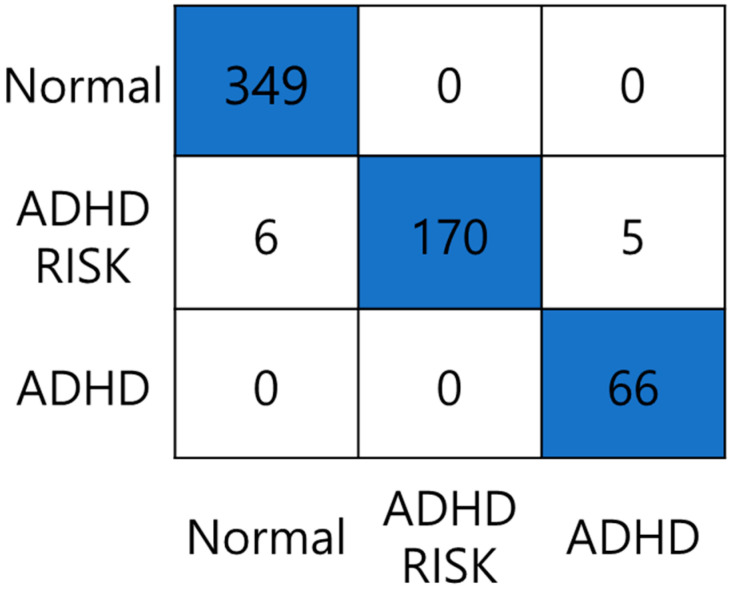
ADHD screening confusion matrix through a deep learning model with bidirectional LSTM and channel attention.

**Figure 9 sensors-23-00278-f009:**
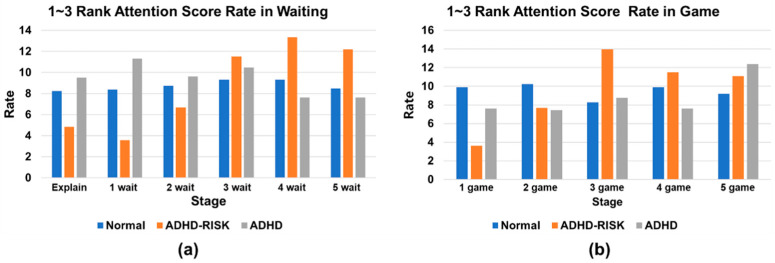
Graph of the distribution rate of the top three attention scores for the three classes for each stage. (**a**) Waiting. (**b**) Game.

**Table 1 sensors-23-00278-t001:** Models and accuracy of studies related to ADHD classification using various datasets.

Dataset	Year	Model	Accuracy
IMU data [17]	2014	SVM	95.12%
EEG [19]	2021	LSTM	92.2%
EEG [22]	2022	CNN	97.7%
fMRI data set [21]	2020	SVM	88.1%
fMRI data set [23]	2022	CNN	73.73%
fMRI data set [24]	2022	CNN	69%
fMRI data set [25]	2022	Spatiotemporal attention auto encoder	72.5%
test scores [18]	2020	Random forest	87%
test scores [18]	2020	SVM	86.3%
test scores [26]	2022	Decision tree	92.5%

**Table 2 sensors-23-00278-t002:** Total number of children who participated in this study and the number of children in each class.

Class	Number of Participants
ADHD	66
ADHD at risk	181
Normal	349
Total	596

**Table 3 sensors-23-00278-t003:** ADHD screening results through a deep learning model applied with bidirectional LSTM and channel attention.

Label	*Acc* (%)	*Se* (%)	*Sp* (%)	*F*1 (%)	*FPR*	*FDR*	*FNR*
Normal	98.99	100	97.57	99.14	0.024	0.016	0
ADHD–RISK	98.15	93.92	100	96.86	0	0	0.06
ADHD	99.16	100	99.06	96.35	0.009	0.07	0

**Table 4 sensors-23-00278-t004:** Average attention score for each stage of all children obtained through the channel attention layer.

	Explain	1 Wait	2 Wait	3 Wait	4 Wait	5 Wait	1 Game	2 Game	3 Game	4 Game	5 Game
**Average (%)**	8.808	9.146	8.896	9.201	9.240	8.821	9.116	9.437	8.945	9.273	9.119

**Table 5 sensors-23-00278-t005:** Distribution rate of the top three attention scores for each stage and class of all children acquired through the channel attention layer.

(%)	Explain	1 Wait	2 Wait	3 Wait	4 Wait	5 Wait	1 Game	2 Game	3 Game	4 Game	5 Game
Normal	8.244	8.363	8.722	9.319	9.319	8.483	9.916	10.231	8.288	9.916	9.2
ADHD–RISK	4.848	3.561	6.667	11.515	13.333	12.197	3.636	7.697	13.97	11.485	11.091
ADHD	9.524	11.333	9.619	10.476	7.619	7.619	7.619	7.429	8.762	7.619	12.381

## Data Availability

Not applicable.

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
