# Peer review of "Deep Learning-Based ADHD and ADHD-RISK Classification Technology through the Recognition of Children’s Abnormal Behaviors during the Robot-Led ADHD Screening Game"

_sensors, 2022, doi:10.3390/s23010278_

Round 1

Reviewer 1 Report

1) The work lacks a comparison to the previous works. Include some numerical comparisons.

2) Write the novelty and contributions of the work in bullets.

3) State clearly why the problem under consideration needs deep learning as very complex algorithms.

4) Provide complexity analysis of the proposed algorithm   

5) fix equation 4. 

6) Provide more discussion about figure 9. 

Reviewer 2 Report

1. The abstract should be revised. It should contain the following information: introduction, problems statement, aim/objectives, methodology, findings, the significance of findings and a concluding statement.

2. The author's contribution should be highlighted.

3. The paper's organisation should be included at the end of the Introduction section.

4. Previous studies are weak. Compare this study with previous studies in a table. How is this study different from other studies?

5. Many grammatical mistakes are found throughout the text. Proofreading is needed for this manuscript.

6. Most of the references are old. Include some recent references from 2022.

7. what is the reference of equation 1?

8. Compare the findings with the existing studies.

Reviewer 3 Report

I have reviewed in detail the work you have done, titled “Deep learning-based ADHD and ADHD-RISK classification technology through the recognition of children's abnormal behaviors during the robot-led ADHD screening game”. The points that I think are missing are listed below.

The model proposed in the summary should be highlighted. A paragraph about the organization of the article should be added at the end of the Introduction section.The performance measurement metrics in Table 2 should be expanded. I recommend you review the related article https://doi.org/10.1016/j.cmpb.2021.106369.The contributions of the study should be highlighted in the introduction. The study should be compared with similar studies in the literature. The advantages of the study should be highlighted. Figures should be reviewed. The texts in the figures are not readable. It would be more appropriate to write the texts in Figure 4 in lowercase.

Round 2

Reviewer 1 Report

The authors need to implement my comments in the manuscript, not only in the response report. For example, they do not includes bullets for contributions in the paper.  

Author Response

We have added the contents of the previous review to the text. thank you

Reviewer 2 Report

The authors of this manuscript have revised this manuscript according to my previous comments. However, the previous studies are still weak. Previous studies should be classified according to their findings and How this study differs from previous studies.

Author Response

I wrote the review content in the attached file. thank you
